# Hypotensive Effect of Electric Stimulation of Caudal Ventrolateral Medulla in Freely Moving Rats

**DOI:** 10.3390/medicina59061046

**Published:** 2023-05-29

**Authors:** Žilvinas Chomanskis, Vytautas Jonkus, Tadas Danielius, Tomas Paulauskas, Monika Orvydaitė, Kazimieras Melaika, Osvaldas Rukšėnas, Vaiva Hendrixson, Saulius Ročka

**Affiliations:** 1Clinic of Neurology and Neurosurgery, Faculty of Medicine, Vilnius University, LT-01513 Vilnius, Lithuania; 2Faculty of Physics, Vilnius University, LT-01513 Vilnius, Lithuania; 3Institute of Applied Mathematics, Faculty of Mathematics and Informatics, Vilnius University, LT-01513 Vilnius, Lithuania; 4Department of Neurobiology and Biophysics, Institute of Biosciences, Life Sciences Center, Vilnius University, LT-01513 Vilnius, Lithuania; 5Faculty of Medicine, Vilnius University, LT-01513 Vilnius, Lithuania; 6Department of Physiology, Biochemistry, Microbiology and Laboratory Medicine, Institute of Biomedical Sciences, Faculty of Medicine, Vilnius University, LT-01513 Vilnius, Lithuania

**Keywords:** blood pressure, caudal ventrolateral medulla, electric stimulation, freely moving rat

## Abstract

*Background and Objectives*: An altered sympathetic function is established in primary arterial hypertension (PAH) development. Therefore, PAH could be targeted by applying an electric current to the medulla where reflex centers for blood pressure control reside. This study aims to evaluate the electric caudal ventrolateral medulla (CVLM) stimulation effect on blood pressure and animal survivability in a freely moving rat model. *Materials and Methods*: A total of 20 Wistar rats aged 12–16 weeks were randomly assigned to either: the experimental group (*n* = 10; electrode tip implanted in CVLM region) or the control group (*n* = 10; tip implanted 4 mm above the CVLM in the cerebellum). After a period of recovery (4 days), an experimental phase ensued, divided into an “OFF stimulation” period (5–7 days post-surgery) and an “ON stimulation” period (8–14 days post-surgery). *Results*: Three animals (15%, one in the control, two in the experimental group) dropped out due to postoperative complications. Arterial pressure in the experimental group rats during the “OFF stimulation” period decreased by 8.23 mm Hg (*p* = 0.001) and heart rate by 26.93 beats/min (*p* = 0.008). *Conclusions*: From a physiological perspective, CVLM could be an effective deep brain stimulation (DBS) target for drug-resistant hypertension: able to influence the baroreflex arc directly, having no known direct integrative or neuroendocrine function. Targeting the baroreflex regulatory center, but not its sensory or effector parts, could lead to a more predictable effect and stability of the control system. Although targeting neural centers in the medullary region is considered dangerous and prone to complications, it could open a new vista for deep brain stimulation therapy. A possible change in electrode design would be required to apply CVLM DBS in clinical trials in the future.

## 1. Introduction

Deep brain stimulation (DBS) is a modern therapy used to treat symptoms of patients suffering from Parkinson’s disease, essential tremor, and dystonia [1]. Applying an electric current to the nervous tissue is attempted for various conditions in many preclinical and clinical trials [2,3,4,5,6]. However, for the last 20 years, the indications of DBS have failed to expand significantly. Major depression, drug-resistant hypertension, obesity, and various disturbances of consciousness, although very tempting, are still not amenable to treatment by DBS in the clinical setting [7,8,9,10].

One disorder that could be treated by applying an electric current to the brain tissue is drug-resistant arterial hypertension [11]. Primary arterial hypertension (PAH) is a major risk factor for coronary heart disease, heart failure, or stroke, accounting for 13% of deaths globally [12,13]. Although PAH is a multifactorial disease, sympathetic hyperactivity is clearly embedded in the pathophysiology of drug-resistant hypertension [14]. There is considerable evidence to suggest the possible effectiveness of DBS in treating drug-resistant hypertension [15,16,17,18,19,20,21]. Up to 44% of PAH could be due to increased sympathetic nervous system tone, as evidenced by studies assessing the norepinephrine spillover in urine or the concentration of catecholamines in plasma [15]. One of the most effective groups of antihypertensive drugs directly targets the central portion of the sympathetic nervous system [16]. Salt-sensitive and obesity-induced hypertension are associated with increased sympathetic nervous system activity [17,18,19]. Successful interventional therapies for drug-resistant hypertension, such as baroreflex stimulation (BS) or renal artery denervation (RAD) procedures, deal with sensory or effector parts of the autonomic nervous system [20,21].

Several DBS targets are used in preclinical research and are considered suitable candidates for hypertension [22]. The best-defined target—periaqueductal grey matter (PAG)—is a coalescence of grey matter columns that integrates information from other autonomic nervous system centers [23]. In addition to participating in blood pressure maintenance, PAG has some nociceptive properties and engages in “fight or flight” reactions. All the scarce clinical evidence we have about DBS efficacy in lowering blood pressure was gained from a few cases reporting the placement of electrodes in ventral PAG [24]. Other possible targets reported in the literature are the rostral subcallosal neocortex, subthalamic nucleus, posterior hypothalamus, orbitofrontal cortex, and insular cortex [22]. By not being reflexive but rather integrative centers, these targets share the same inherent problems. Their hypotensive effect should be mild, as demonstrated by clinical reports on PAG, and could lead to unwanted neuroendocrine or behavioral side effects.

Applying electric current to the caudal ventrolateral medulla (CVLM) in non-survival surgery leads to profound hypotensive effects [25]. CVLM is a major relay center of the baroreflex arc that directly inhibits the rostral ventrolateral medulla (RVLM) [26]. Long-term CVLM electric stimulation in awake and freely moving rats was not reported in the literature. Lower medullary implantations are believed to be dangerous and prone to severe neurological complications [27]. However, a successful survival surgery and stimulation, as low as the RVLM, were attempted for other than hemodynamic reasons [28]. Our previous study demonstrated considerable survivability of rats after electrode implantations in the CVLM region [29]. However, the article dealt primarily with methodological issues of the brain stimulator, and only modest evidence of the effectiveness of CVLM DBS was demonstrated.

This study aims to evaluate the electric CVLM stimulation effect on blood pressure and animal survivability in a freely moving rat model.

## 2. Materials and Methods

### 2.1. Design of the Device

A backpack-mounted device (combined brain stimulator and pressure sensor) was custom-built from the “off-the-shelf” components allowing wireless data transmission (Figure 1) [29]. Wireless transmission of blood pressure data to the nearby computer is based on the low-cost microcontroller (ESP8266 chip; Espressif Systems, Shanghai, China). Blood pressure measurements and brain stimulations were controlled by a microcontroller (LPC845; NXP Semiconductors, Eindhoven, The Netherlands). Voltage pulse generation was performed by an LPC845 and a digital to analogue converter (MCP4921; Microchip Technology Inc., Chandler, AZ, USA). The UART interface was used for data and command exchange between the ESP8266 and LPC845. A sensitive and stable pressure sensor BPS130 (Bourns, Inc., Riverside, CA, USA) was chosen with an upper pressure range limit exceeding 300 mm Hg [30]. The created device can generate a pulse of arbitrary shape with pulse amplitude, pulse duration, and pulse repetition ranges in 0–140 µA, 40–300 µs, and 2–500 ms, respectively. The built device, although not attempted, could be programmed to work in closed-loop stimulation design.

### 2.2. Preoperative Period

All study protocols and experiments followed current European Union regulations. Lithuanian institutional authorities approved the experiments—State Food and Veterinary Service, No. G2-128. A total of 20 Wistar male rats aged 12–16 weeks weighing 320–421 g were randomly assigned to experimental (*n* = 10) or control groups (*n* = 10). The rats were housed individually with 12 h light/dark cycles, in constant temperature conditions, and food and drinks were allowed ad libitum. One week before the planned surgery, animals were equipped with a jacket and a case simulating the device size and weight to seek adaptation.

### 2.3. Surgery

The animal’s body temperature was maintained at 36.6 °C during the surgery with a thermostatically controlled heating pad. Under sevoflurane (Baxter) anesthesia (3–4% in a mixture of 100% O_2_ administered through a nose cone), rats were placed in a supine position and a paramedian cervical incision was made. The left carotid artery was dissected under a light microscope (Leica Microsystems GmbH, Wetzlar, Germany) with 6.4× magnification and the vagal nerve was carefully separated and spared. The artery was cannulated with a polyethylene tube, 0.96 mm external diameter, filled with heparinized (500 UI/mL) 99.5% glycerol solution (Smiths Medical International, Ltd., Luton, UK). The catheter was tunneled under the skin, externalized over the suprascapular region, and connected to a device for blood pressure recording. Afterwards, animals were placed in a stereotactic frame (Narishige Scientific Instrument Lab., Tokyo, Japan) for electrode implantation. A custom-made stainless steel twisted bipolar electrode (Goodfellow Cambridge, Ltd., Huntingdon, UK) with polyamide insulation was inserted in the right CVLM. The coordinates of CVLM were: −4.40 mm anteroposteriorly, 2.10 mm mediolaterally, and 9.90 mm dorsoventrally from lambda, according to Paxinos and Goodchild [31,32]. The implantation site was verified by two consecutive stimulation trains of 3 s (100 μA, 50 Hz, 0.1 ms) that caused an arterial blood pressure decrease of at least 20 mm Hg. After the accurate placement of the electrode, a head post made of dental micro-hybrid composite FlowX (ORBIS Dental Handelsgesellschaft mbH., Munster, Germany) was affixed.

Identical procedures were carried out in both groups of rats except for the final positioning of the electrodes, which in the control group of rats was 4 mm above the intended CVLM target. Test stimulation during electrode insertion in this group had no hemodynamic response.

### 2.4. Postoperative Period and Data Analysis

The postoperative and data collection period lasted continuously 24 h per day for two weeks. Catheter patency was maintained by replacing lock solutions (500 IU heparin/99.7% glycerol) once per day with the known filling volume of the system. Continuous blood pressure recordings were taken with no additional infusions of heparinized glycerol lock solution. Only rats free of neurological compromise were enrolled in the final calculations of blood pressure data.

The experimental period was divided into three parts: the recovery phase (days 1–4), the “OFF stimulation” phase (days 5–7), and the “ON stimulation” phase (days 8–14). The recovery phase, during which no stimulation was attempted, allowed complete healing after surgery and stabilization of the blood pressure curve. The data were continuously collected during the “OFF stimulation” period, but no stimulation was attempted. During the “ON stimulation” phase, continuous stimulation was performed with parameters 50 μA, 50 Hz, and 0.1 ms.

Blood pressure data were transmitted to a nearby PC running a Linux-based operating system with a PostgreSQL database. R and MS Excel were used for the analysis of the blood pressure data. The heart rate was extracted by smoothing data using Gaussian kernels with 10 kernel bandwidth.

Mean blood pressure and average heart rate differences between the “OFF stimulation” and “ON stimulation” phases were compared by dependent Student *t*-test in both groups of rats. An independent Student *t*-test was used to compare mean blood pressure and average heart rate differences between the groups during the “OFF stimulation” and “ON stimulation” periods. Statistical significance was taken at *p* < 0.05. Data were expressed as the mean ± standard deviation of the mean.

Animals of our Wistar rats breeding colony typically show mean blood pressure ranging from 85 to 105 mm Hg and average heart rates from 340 to 420 beats *p*/min. The pressure range could, of course, depend on various factors, such as the dampening effect of the catheter system, the lock solutions used, etc.

## 3. Results

### 3.1. Survivability and Neurological Compromise

Rats in both groups that responded to stimulation by an observable decrease in blood pressure (responder), did not respond to stimulation (non-responder), died during the experiment (death), or had neurological side effects (neurological symptoms) are presented in Figure 2. As expected, no rats hemodynamically responded to stimulation in the control group. Six rats (60%) responded to stimulation with a noticeable decrease in blood pressure in the experimental group (Figure 3). One rat in the experimental group died 6 h after the surgery with an evident hypertensive crisis. An autopsy revealed a subdural hematoma around both cerebellar hemispheres and a small hemorrhagic lesion at the electrode tip in the CVLM region. Two rats per both groups had transitory neurological side effects: right limb ataxia attributable to the right cerebellar lesion. Although they improved after a few days, both rats were eliminated from the final hemodynamic data analysis. An autopsy revealed no hemorrhagic lesions at the electrode tip or around the cerebellar hemispheres. During the stimulation phase, no observable change in behavioral or neurological functions were noted in both groups of rats.

### 3.2. Hemodynamic Data

Typical changes in mean blood pressure and average heart rate during ten days in a single responder rat of the experimental group are presented in Figure 3. Blood pressure decreased throughout the first day of the “ON stimulation” period and plateaued afterwards. The change in heart rate was not so noticeable; however, in this particular example, a trend towards increasing heart rate variability and more pronounced bradycardic episodes during rest periods is evident. A comparison of mean blood pressure and average heart rate in “OFF stimulation” and “ON stimulation” periods in both groups is shown in Figure 4. 

The “OFF stimulation” baseline mean arterial pressure was 96.22 ± 5.79 mm Hg and 90.84 ± 5.55 mm Hg in rats of experimental and control groups, respectively. The difference in blood pressure during the “OFF stimulation” period between groups, as expected, was not significant (*p* = 0.08). During the “ON stimulation” period, mean arterial pressure decreased by 8.23 mm Hg (*p* = 0.001) in the experimental group and increased by 0.75 (*p* = 0.78) in the control group. The “OFF stimulation” baseline average heart rate was 376.34 ± 15.46 beats/min and 365.94 ± 18.26 beats/min in rats of experimental and control groups, respectively. The difference in average heart rate during the “OFF stimulation” period between groups, as expected, was not significant (*p* = 0.297). During the “ON stimulation” period average heart rate decreased by 26.93 beats/min (*p* = 0.008) in the experimental group and by 3.10 (*p* = 0.503) in the control group. There was no statistically significant difference between the two groups during the “ON stimulation” periods in MAP (*p* = 0.229) and BP (*p* = 0.139).

## 4. Discussion

CVLM is the primary inhibitory nucleus of the brainstem circuitry responsible for the reflex control of arterial blood pressure [33]. Baroreceptor afferents synapse on a subpopulation of neurons in the nucleus of the solitary tract (NTS); these neurons, in turn, excite GABAergic neurons in the CVLM. CVLM inhibits rostral ventrolateral medulla (RVLM) neurons, which send direct projections to the intermediolateral column in the spinal cord, where preganglionic sympathetic neurons reside. CVLM and, in particular, RVLM have vast afferent and efferent connections with other centers of sympathetic control, such as medullary respiratory neurons, medullary lateral tegmental field, midline raphé nuclei, A5 cell group, parabrachial nuclei, midbrain periaqueductal gray, lateral and periventricular nuclei of the hypothalamus, and prefrontal cortex [34]. RVLM plays a pivotal role in the regulation of renal sympathetic nerve activity, influencing humoral mechanisms of the renin–angiotensin–aldosterone system [35].

In the past, placing electrodes or microcannulas in the CVLM region was confined to only acute experiments [36,37]. It was thought that constant motion of the mobile part of the brainstem during animal movements would lead to severe damage around the implanted electrode tip rendering chronic experimentation impossible [27]. This study revealed that CVLM implantation procedures do not lead to significant mortality in animals. The rate of neurological complications is no different than after electrode implantations in the cerebellar region. One loss of an animal that could be linked to CVLM damage should not discourage repeating similar experiments in the future. Furthermore, the lesion in the CVLM region could have been secondary to a massive subdural hematoma around the cerebellum, causing a significant brainstem shift. The cause of subdural bleeding itself could be attributable to an electrode trajectory being close to a branch of the posterior inferior cerebellar artery running in the paramedian sulcus, an easily avoidable complication.

During acute experiments, the electric or chemical stimulation of the CVLM region leads to a profound decrease in blood pressure, bradycardia, and reduced sympathetic nerve activity [36,37]. Although the hemodynamic change was not so pronounced as in unconscious animals, our study does show effective stimulation that lasted until the end of the experiment. It should be noted that the stimulation effect is not evident at once: it lags behind and takes up to two days to plateau (Figure 3). The lag in effect might explain why O‘Callaghan et al. could not locate a hypotensive response when electrically stimulating PAG for up to 20 min in conscious, unrestrained, spontaneously hypertensive rats [38]. This lag could even support the notion that something other than the acute vasodilatory effect could be causing it. As mentioned, links between the baroreflex arc and the renin–angiotensin–aldosterone system are firmly embedded in the blood pressure regulation circuitry [35]. Although this study was not designed to show the CVLM stimulation effect on the humoral control of blood pressure, this could be an interesting direction for future research.

It is known that some centers of cardiovascular control in the brain overlap with centers of locomotion [39]. In rats, cardiorespiratory and locomotor centers have been identified in the same regions of PAG, posterior hypothalamic area, NTS, RVLM, and the cuneiform nucleus. Although behavioral experiments were not part of our study, we have not observed any change in the activity or behavioral patterns of rats in or between the groups.

As expected, very limited data from clinical studies support the possible role of CVLM as a candidate target for managing drug-resistant hypertension [40,41,42]. Jannetta, a pioneer of microvascular decompressions, proposed the “neurogenic hypertension” theory, which states that pulsatile vascular compression of the ventrolateral medulla could predispose patients to drug-resistant hypertension [41]. In the early 1980s, he successfully treated 32 of 36 patients by translocating vertebral arteries away from the region of the ventrolateral medulla. Another more recent study investigated changes in autonomic activities and systemic circulation generated by surgical manipulation or local electrical stimulation to the human brain stem intraoperatively. Yet another study demonstrated similar results of local surface stimulations on the pontomedullary junction; the authors theorized that electric current spread to RVLM could be responsible for the acute hypertensive effect [42].

The main disadvantage of the CVLM region is its inferior location in the brainstem axis. This renders it almost impossible for current deep brain stimulation (DBS) systems to target this region. In the literature, we found the locus coeruleus region and parabrachial complex, both in the lower pons, being the lowest human brainstem targets where insertion of DBS electrodes was successfully attempted [43,44]. While the position of CVLM could be a main disadvantage, its proximity to the brainstem surface could be beneficial. Coursing just 300 μm below pia matter, CVLM could be stimulated by the current spread from plate-like electrodes without penetrating the medulla directly [45]. The above-mentioned electric stimulation experiments on the brainstem surface support this idea and could be a clue for translational human trials in the future [41,42].

Clinical data evaluating the hypotensive DBS effect in drug-resistant hypertension come from studies targeting PAG [11,24,46,47]. Theoretically, CVLM, although a much more difficult-to-reach target, has some inherent advantages over PAG. Firstly, CVLM has no known direct nociceptive, neuroendocrine, or integrative behavioral function. It is known that stimulation of PAG in animals could lead to intense “fight or flight” reactions and suppression of pain pathways [38,45]. As opposed to PAG, CVLM’s primary function is to relay information from NTS to the RVLM in a reflexive manner [45]. The second theoretical advantage over PAG is CVLM being the lowest inhibitory center in the axis of the central autonomic network. CVLM stimulation should influence different pathogenetic circuitry of hypertension, no matter the cause of resistant hypertension, be it obesity-induced, salt-sensitive, or hypertension caused by sensitization of autonomic pathways [14,29].

RAD and BS procedures are established methods for treating resistant hypertension [20,21]. Although both methods are effective, these techniques are associated with a few inherent disadvantages that could be resolved by DBS [35]. First, there is no simple, reliable, and repeatable method in clinical practice to measure sympathetic nerve activity after the procedure. Therefore, in the early postoperative period, the magnitude of denervation cannot be easily quantified, leading to difficult predictions regarding the procedure’s efficacy. Second, RAD is a destructive and irreversible method with permanent damage to the renal sympathetic nerves. Third, the non-modulating nature of RAD does not allow for easy effect modification. RAD is also characterized by general complications of standard percutaneous arterial interventions: bleeding and hematomas at the puncture site, renal artery dissection, risk of thromboembolic formation, and iatrogenic arterial damage.

Unlike RAD, the BS procedure is non-destructive, and optimization of the device effect is possible by changing the stimulation parameters [35]. However, the procedure is much more expensive than RAD. In addition, the more invasive nature of the BS may lead to higher infectious and implant-related complication rates [48]. While this stimulator cannot be wirelessly recharged, a replacement procedure is required every few years. In addition, BS electrodes are implanted around the carotid arteries, so there is a theoretical risk of thromboembolic stroke due to the migration of atherosclerotic plaques. Moreover, continuous stimulation of baroreceptors may increase the risk of falls that is related to decreased blood pressure adaptation to sudden changes in body position.

The main limitation of this study is the lack of histological validation of the target. Although we used physiological confirmation during the electrode placement, post hoc histological validation would be beneficial to evaluate the histological damage on the CVLM.

Future studies should address the comparison of bilateral and unilateral CVLM stimulation: whether bilateral stimulation would lead to an enhanced hypotensive effect or a higher mortality rate. Furthermore, it would be beneficial to substitute the current model of Wistar rats with spontaneously hypertensive rats, as a more pronounced hypotensive effect should be expected.

## 5. Conclusions

Although targeting neural centers in the medullary region is considered dangerous and prone to complications, it could open a new vista for deep brain stimulation therapy. This study shows that CVLM implantation procedures do not lead to significant mortality in animals, and the rate of neurological complications is acceptable. Furthermore, this study indicates that chronic stimulation of the CVLM region can effectively lower arterial blood pressure in otherwise healthy conscious Wistar rats. As discussed, stimulation of the CVLM region could have some inherent benefits as opposed to PAG stimulation. With its intricate cardiovascular, respiratory, wakefulness, and other regulatory networks, the lower medullary region could be an effective yet challenging site for electrode implantation. However, it could still help treat such complex diseases and conditions as drug-resistant hypertension, vasovagal syncope, or even vegetative state.

Considering the high prevalence and frequent drug resistance of PAH and the significance of the sympathetic nervous system in PAH pathogenesis, it would be beneficial to have more treatment options added to already established techniques. DBS could be a minimally invasive, safe, and non-destructive procedure that acts directly on the regulatory centers of blood pressure control.

## Figures and Tables

**Figure 1 medicina-59-01046-f001:**
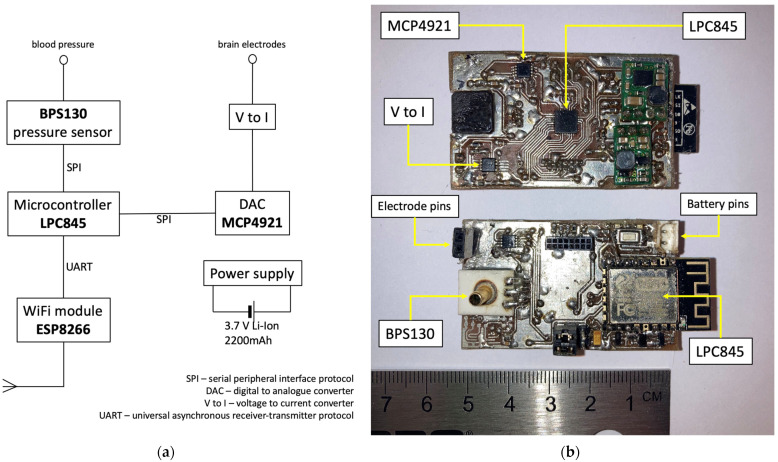
The basic design of the device (**a**) and two working models (**b**) with arrows pointing to critical elements mentioned in (**a**).

**Figure 2 medicina-59-01046-f002:**
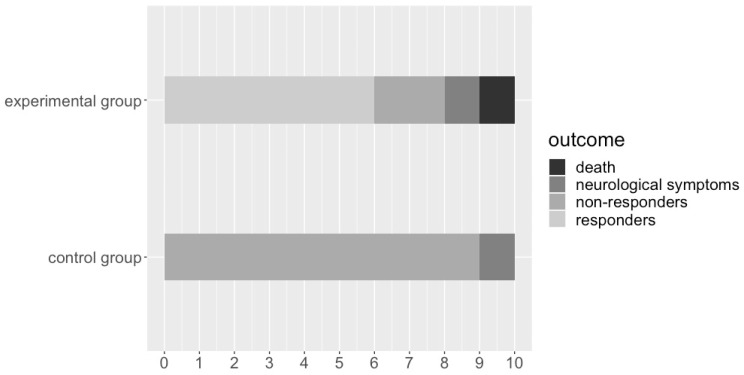
The outcomes of rats in the experimental and control groups.

**Figure 3 medicina-59-01046-f003:**
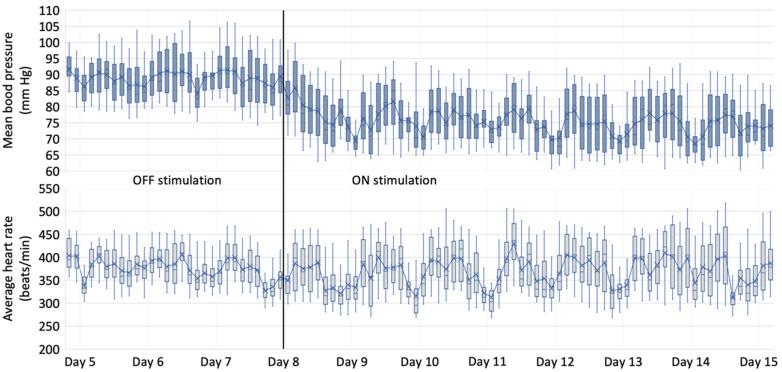
The cardiovascular response in responder Wistar rat of the experimental group evoked by electrical stimulation in the CVLM region. Mean arterial pressure and average heart rate plotted as time series boxplots during “OFF stimulation” and “ON stimulation” periods.

**Figure 4 medicina-59-01046-f004:**
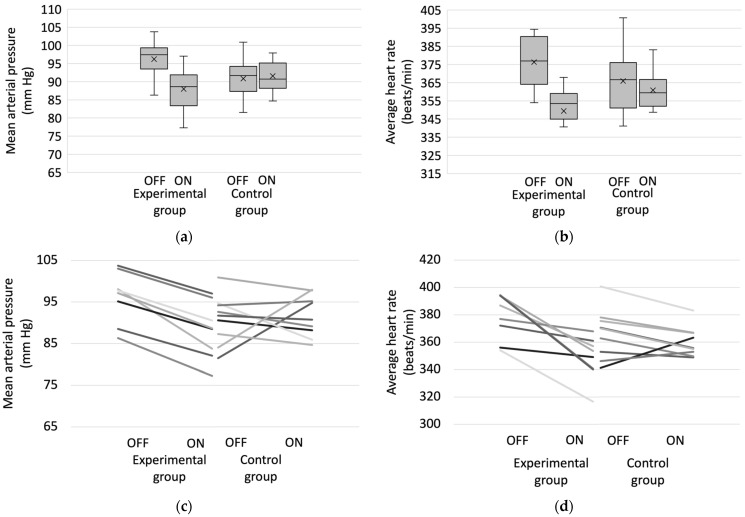
Comparison of mean arterial pressure (**a**) and average heart rate (**b**) during “ON stimulation” and “OFF stimulation” periods in experimental and control groups. The change in mean arterial pressure (**c**) and average heart rate (**d**) in individual Wistar rats in control and experimental groups. OFF—“OFF stimulation” period; ON—“ON stimulation” period.

## Data Availability

The data presented in this study are available on reasonable request from the corresponding author (Ž.C.).

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
