# Peer review of "Hypotensive Effect of Electric Stimulation of Caudal Ventrolateral Medulla in Freely Moving Rats"

_medicina, 2023, doi:10.3390/medicina59061046_

Round 1

Reviewer 1 Report

It is an exciting topic for treating Hypertension by stimulating the caudal ventrolateral medulla. A lot of new tools for the treatment of hypertension were discussed last years. One of them is renal denervation and stimulation of ganglion. Auotors used another approach to regulate blood press by neurostimulation. The experiment was perfectly designed and showed positive results. The authors view the new approaches to the treatment of resistant hypertension without comparing them with different interventional and surgical methods. In the discussion section, I recommend comparing different intervention approaches for hypertension control. The conclusion needs some correction without including information about resistant hypertension. The conclusion section is not for discussion, in my opinion. After the correction, this article could be published.

Author Response

Reviewer 1: “It is an exciting topic for treating Hypertension by stimulating the caudal ventrolateral medulla. A lot of new tools for the treatment of hypertension were discussed last years. One of them is renal denervation and stimulation of ganglion. Authors used another approach to regulate blood press by neurostimulation. The experiment was perfectly designed and showed positive results.”

Response: Thank you for your kind words!

Reviewer 1: “The authors view the new approaches to the treatment of resistant hypertension without comparing them with different interventional and surgical methods. In the discussion section, I recommend comparing different intervention approaches for hypertension control.”

Response: We thank the reviewer for this suggestion. Of course, we can only indulge in physiological theoretical comparisons of CVLM DBS with other interventions for resistant hypertension, as no clinical studies compare these established techniques to date. As suggested by the reviewer, we added more information regarding different interventional approaches for resistant hypertension.

Reviewer 1: “The conclusion needs some correction without including information about resistant hypertension. The conclusion section is not for discussion, in my opinion. After the correction, this article could be published.”

Response: Thank you! We hope that we understood this suggestion as was intended by a dear reviewer. We added some remarks regarding resistant hypertension and DBS that would finalize the conclusions section. In the conclusions section, we summarized our study's main ideas or the so-called “philosophy behind the study”.

Reviewer 2 Report

Deep brain stimulation (DBS) is a modern therapy used to treat symptoms of patients suffering from Parkinson‘s disease, essential tremor, and dystonia. Applying an electric current to the nervous tissue is attempted for various conditions in many preclinical and clinical trials. There is considerable evidence to suggest the possible effectiveness of DBS in treating drug-resistant hypertension. This is a large and highly structured experimental study.

The novelty was the use of long-term electrical stimulation of the CVLM in awake and freely moving rats, which had not been reported in the literature before.

The article is well organized; it includes Introduction, Materials and Methods, Results, Discussion and Conclusions.

               The methodology is clearly described. 

Recommendations:

In subchapter 2.4. I recommend specifying the reference ranges for blood pressure and heart rate.

In the results section I recommend to state whether there are significant differences between the two groups in blood pressure and heart rate (initial and final).

Figure 3 does not specify to which group the graphical representation refers.

Line 267 - specify the unit of measurement.

               The conclusions are consistent with the evidence and arguments presented; they answer the study question.

               The article is well written and easy to understand.

Author Response

Reviewer 2: “Deep brain stimulation (DBS) is a modern therapy used to treat symptoms of patients suffering from Parkinson‘s disease, essential tremor, and dystonia. Applying an electric current to the nervous tissue is attempted for various conditions in many preclinical and clinical trials. There is considerable evidence to suggest the possible effectiveness of DBS in treating drug-resistant hypertension. This is a large and highly structured experimental study.

The novelty was the use of long-term electrical stimulation of the CVLM in awake and freely moving rats, which had not been reported in the literature before.

The article is well organized; it includes Introduction, Materials and Methods, Results, Discussion and Conclusions. The methodology is clearly described.”

Response: Thank you!

Reviewer 2: “In subchapter 2.4. I recommend specifying the reference ranges for blood pressure and heart rate.”

Response: We hope that we understood the reviewer's recommendation correctly. By “specifying the reference ranges for blood pressure and heart rate” we believe the reviewer wanted us to mention the normal MAP and BP ranges of typical Wistar rats used in the experiment. We added this information regarding reference ranges of otherwise healthy Wistar rats of our in-house breeding colony in the text.

Reviewer 2: “In the results section I recommend to state whether there are significant differences between the two groups in blood pressure and heart rate (initial and final).”

Response: Thank you for this suggestion. We added this information in the results section.

Reviewer 2: “Figure 3 does not specify to which group the graphical representation refers.”

Response: Thank you for pointing this out. It was corrected as suggested.

Reviewer 2: “Line 267 - specify the unit of measurement.”

Response: We corrected this mistake; somehow, the micro symbol is notorious for behaving mischievously in the main text.

Reviewer 2: “The conclusions are consistent with the evidence and arguments presented; they answer the study question. The article is well written and easy to understand.”

Response: We are very grateful for the reviewer’s positive remarks. Thank you!